

# An ensemble of AMIP simulations with prescribed land surface temperatures

Duncan Ackerley[1,2], Robin Chadwick[3], Dietmar Dommenget[1], and Paola Petrelli[4]

[1]ARC Centre of Excellence for Climate System Science, School of Earth Atmosphere and Environment, Monash University, Clayton 3800, Victoria, Australia
[2]Met Office, Exeter, UK
[3]Met Office Hadley Centre, Exeter, UK
[4]ARC Centre of Excellence for Climate System Science, Institute for Marine and Antarctic Studies, University of Tasmania, Hobart, Tasmania, Australia

**Correspondence:** Duncan Ackerley (duncan.ackerley@metoffice.gov.uk)

**Abstract.** General circulation models (GCMs) are routinely run under Atmospheric Modelling Intercomparison Project (AMIP) conditions with prescribed sea surface temperatures (SSTs) and sea ice concentrations (SICs) from observations. These AMIP simulations are often used to evaluate the role of the land and/or atmosphere in causing the development of systematic errors in such GCMs. Extensions to the original AMIP experiment have also been developed to evaluate the response of the global cli-

5  mate to increased SSTs (prescribed) and carbon-dioxide ($CO_2$) as part of the Cloud Feedback Model Intercomparison Project (CFMIP). None of these international modelling initiatives has undertaken a set of experiments where the land conditions are also prescribed, which is the focus of the work presented in this paper. Experiments are performed initially with freely-varying land conditions (surface temperature and, soil temperature and mositure) under five different configurations (AMIP, AMIP with uniform 4 K added to SSTs, AMIP SST with quadrupled $CO_2$, AMIP SST and quadrupled $CO_2$ without the plant stomata

10  response, and increasing the solar constant by 3.3%). Then, the land surface temperatures from the free-land experiments are used to perform a set of "AMIP-prescribed land" (PL) simulations, which are evaluated against their free-land counterparts. The PL simulations agree well with the free-land experiments, which indicates that the land surface is prescribed in a way that is consistent with the original free-land configuration. Further experiments are also performed with different combinations of SSTs, $CO_2$ concentrations, solar constant and land conditions. For example, SST and land conditions are used from the AMIP

15  simulation with quadrupled $CO_2$ in order to simulate the atmospheric response to increased $CO_2$ concentrations without the surface temperature changing. The results of all these experiments have been made publicly available for further analysis. The main aims of this paper are to provide a description of the method used and an initial validation of these AMIP-prescribed land experiments.



## 1   Introduction

In order to evaluate the atmosphere and land modules of general circulation models (GCMs), simulations can be run under "Atmospheric Model Intercomparison Project" (AMIP) specifications (Gates, 1992; Gates et al., 1999). Typically, both sea surface temperatures (SSTs) and sea ice concentrations (SICs) are prescribed from observations over some reference period (e.g. 1979–2014 in the Coupled Model Intercomparison Project Phase 6—CMIP6—experiment, see Eyring et al., 2016) with the atmosphere and land allowed to respond freely to the SST and SIC field. Such AMIP simulations help to understand the role of the atmosphere and/or land in the development of model errors. Further to the standard AMIP experiments, quadrupled $CO_2$ (amip4xCO2) and spatially uniform 4K SST increase (amip4K) experiments were incorporated as part of CMIP5 (see Taylor et al., 2012) by the Cloud Feedback Model Intercomparison Project (CFMIP, Bony et al., 2011). The amip4xCO2 experiment was designed to identify the "rapid cloud response" to increased $CO_2$ and the amip4K experiment was intended to investigate the impact of the dynamical response of the atmosphere (to the higher SST) on cloud feedbacks (Bony et al., 2011). The CFMIP experiments have also been used to examine the regional precipitation response to both $CO_2$ forcing and higher SSTs (e.g. Bony et al., 2013; Chadwick et al., 2014; He and Soden, 2015). The amip4xCO2 and amip4K experiments are also included in CMIP6 (see Webb et al., 2017). While the AMIP experiments described above are designed to investigate the response of the land and the atmosphere to the imposed SST and $CO_2$ conditions, there is scope to further isolate the response of the atmosphere by prescribing the land conditions too. Such a method of prescribing the land has not been attempted (to our knowledge) as part of the CFMIP/CMIP initiative; however, there are several key issues from the CFMIP and CMIP6 experiments that could at least be partially addressed by running a set of AMIP simulations with prescribed land conditions, for example:

(1) How does the Earth system respond to forcing and what is the role of the land in that response? (Adapted from Eyring et al., 2016)

(2) How can the understanding of circulation and regional scale precipitation (particularly over the land) be improved? (Adapted from Webb et al., 2017)

Prescribing global surface temperatures (including the land) in order to, for example, suppress the surface response to a radiative forcing is not a new idea. Such an approach has previously been used to understand the strength of coupling between the land and atmosphere in GCMs (Koster et al., 2002). In another example, Shine et al. (2003) prescribed land temperatures in order to estimate the climate sensitivity parameter of an intermediate complexity GCM in a variety of greenhouse gas and aerosol forcing experiments. Furthermore, a better estimate of the radiative forcing from e.g. quadrupling $CO_2$ may be attained from GCMs by fixing land surface temperatures as the changes in land temperature can change the atmosphere (e.g. circulation, clouds and precipitation) in a manner that can affect the simulated global radiation balance (Andrews et al., 2012a, 2015). Unfortunately, the method of prescribing land temperatures (as well as SSTs) has not be developed widely for use in



multinational modelling efforts (such as CMIP) and has only been used in one-off idealised modelling experiments such as those described by Dommenget (2009) and Ackerley and Dommenget (2016).

Work by Bayr and Dommenget (2013) used the prescribed land temperature experiments from Dommenget (2009) and data from the CMIP3 experiment to show that higher land temperatures (and specifically increasing the land-sea thermal contrast) is

an important driver of circulation change under global warming. However, there are many different mechanisms/forcing agents that can cause the land surface temperatures to increase (or decrease), which may also have an impact on the global circulation. For example, land surface temperatures increase by more than 4 K in amip4k-type experiments (e.g. Joshi et al., 2008), which indicates that land temperatures can change substantially in response to changes in SSTs. Land temperatures also increase directly in response to increased $CO_2$ concentrations, which cause increased downwelling long-wave radiation and cloud

adjustments (Dong et al., 2009; Cao et al., 2012). This increase in land temperatures forms part of the direct $CO_2$ effect, which drives both global (Allen and Ingram, 2002) and regional (Bony et al., 2013; Chadwick et al., 2014; Merlis, 2015) precipitation responses; however it is currently unclear how much of this effect is due to increases in atmosphere or land temperatures. To complicate matters further over the land, the degree of land surface warming and precipitation change are also sensitive to the physiological response of plant stomata, which close as $CO_2$ concentrations increase and thereby reduce evapotranspiration

and precipitation locally (Doutriaux-Boucher et al., 2009; Boucher et al., 2009; Andrews et al., 2011). Finally, land surface temperatures (and therefore circulation and precipitation) also respond to changes in insolation (e.g. the "abrupt Solar-fixed SST" experiments in Chadwick et al., 2014; Andrews et al., 2012b). Given that all of the different forcing agents outlined above have very different impacts on land temperatures and the global circulation (and precipitation), it would be useful to quantify the separate contributions of the land (temperature and soil moisture), the atmosphere (e.g. increased long-wave absorption),

plant physiology and SSTs to the circulation change separately (and any other aspects of regional and global climate change). Prescribed land experiments could achieve this and the modelling framework developed by Ackerley and Dommenget (2016) for the Australian Community Climate and Earth System Simulator (ACCESS) provides an opportunity to do so. There is also scope to provide a platform to share the results with the wider scientific community through the Australian National Computing Infrastructure (NCI) and the ARC Centre of Excellence for Climate System Science (ARCCSS).

The main aim of this study is to describe and validate a set of AMIP simulations run with freely varying land conditions against those with prescribed land conditions and observational datasets. This study also presents an evaluation of further experiments that employ different combinations of land conditions with the different SST, $CO_2$ and insolation specifications. Finally (and most importantly), the study provides information on where these data can be accessed for others to use.

The model used, experimental outline and reference datasets are given in Section 2, including a description of how the

land datasets were generated. In Section 3, the AMIP simulations with prescribed land are then validated against the original AMIP (freely varying land) simulations from which the land conditions were taken. The results of the AMIP simulations with different combinations of land conditions, SSTs, $CO_2$ concentrations and the solar constant are described in Section 4.1. The results of uniformly increasing the land surface temperatures alone by 4 K and, raising both the land surface and sea surface temperatures by 4 K are discussed in Section 4.2. The summary, concluding remarks and future work (e.g. further development

opportunities) are given in Section 5.





## 2 Model, experiments and reference datasets

### 2.1 Model

#### 2.1.1 General background

The GCM used in this study is the Australian Community Climate and Earth System Simulator (primarily ACCESS1.0) in an
atmosphere-only configuration, which is identical to that used in Ackerley and Dommenget (2016). The version of ACCESS1.0
used here has a horizontal grid spacing of 3.75° (longitude) x 2.5° (latitude) and 38 vertical levels. Parameterized processes
include precipitation, cloud, convection, radiative transfer, boundary layer processes and aerosols. The representation of the
land surface and soil processes is of primary relevance to this study, which is simulated by the Met Office Surface Exchange
Scheme (MOSES, Cox et al., 1999; Essery et al., 2001). Sub-grid scale surface heterogeneity is represented by splitting the
grid box into smaller 'tiles' of which there are nine different types specified. Tiles may be vegetated (e.g. grasses) or non-
vegetated (e.g. bare soil) and the tiles within a grid box can comprise any fractional combination of the surface types. Surface
variables (such as temperature, long-wave and short-wave radiation, and latent and sensible heat fluxes) are calculated for each
tile individually and then summed to give a representative grid box mean value, which is passed back into the main model.
Also of relevance is the representation of soil properties (i.e. soil moisture and temperature), which is simulated over four
vertical layers (0.1, 0.25, 0.65 and 2 m deep). The model code is available by following the instructions in the *Code and data
availability* Section.

#### 2.1.2 Prescribing land temperatures

The land surface temperatures are prescribed using the same method described in Ackerley and Dommenget (2016)—the
reader is directed there for more in-depth discussion. Nevertheless, the calculation of the surface temperature in the free land
simulations (i.e. land surface temperature and, soil moisture and temperature are allowed to vary freely) and the code changes
made to prescribe it are discussed here. An initial estimate of the land surface temperature is calculated from the existing
surface conditions using:

$$T_* = T_s + \frac{1}{A_*} \left[ R_s - H - \lambda E + \frac{C_c}{\Delta t} \left( T_*^{prev} - T_s \right) \right] \tag{1}$$

where the temperature of the first soil layer from the previous time step is denoted as $T_s$ (K), $A_*$ is the coefficient for
converting fluxes into temperature in this instance (W m$^{-2}$ K$^{-1}$), $R_s$ is the net radiation into the surface (both long-wave and
short-wave, W m$^{-2}$), H is the surface sensible heat flux (W m$^{-2}$), $\lambda E$ is the latent heat flux (W m$^{-2}$), $C_c$ is the aereal heat
capacity of the surface (J m$^{-2}$ K$^{-1}$), $\Delta t$ is the length of the time step (s) and $T_*^{prev}$ is the surface temperature from the time
step before the current time (K). The value of $T_*$ from Eq. 1 is then adjusted implicitly within the model depending upon the
moisture availability and changes of state such that:

$$\Delta T_{*_{EVAP}} = -\frac{\Delta H + \Delta(\lambda E)}{A_*} \tag{2}$$





$$T_* = T_{*Eq.1} + \Delta T_{*EVAP} \tag{3}$$

A land surface temperature increment due to evaporation (Eq. 2—$\Delta T_{*EVAP}$, K) is calculated from the adjustments to the sensible heat flux ($\Delta H$, W m$^{-2}$) and the latent heat flux ($\Delta(\lambda E)$, W m$^{-2}$) that are made after diagnosing the moisture availability.

The temperature increment is then simply added to the value of $T_*$ calculated in Eq. 1 (i.e. $T_{*Eq.1}$, K) via Eq. 3. If there is no snow present then $T_*$ is unchanged for the rest of the time step at that land point. If however, snow is present on the land surface then the temperature is adjusted further to account for any snow melt ($\Delta T_{*MLT}$, K) and is again simply added to the value calculated in Eq. 3 by the following:

$$T_* = T_{*Eq.3} + \Delta T_{*MLT} \tag{4}$$

More details on these equations (i.e. Eqs. 1–4) can be found in the relevant papers that describe the MOSES module (i.e. Essery et al., 2001; Cox et al., 1999).

When the surface temperatures are prescribed, Eq. 1 is simply changed to be:

$$T_* = T_{PRES} \tag{5}$$

Where $T_{PRES}$ is the input, prescribed temperature (K) field (discussed in Section 2.2.2, below). Furthermore, the increments

calculated in Eqs. 2–4 are set to zero so that the surface temperature cannot change implicitly within the time step. The surface radiation budget therefore only depends upon $T_{PRES}$.

It is also worth noting here that the existing ACCESS model code has the option for prescribing deep soil temperatures and soil moisture content. When the soil temperatures and moisture are prescribed (as stated in the experiments below), that option is switched on in the code and soil moisture and deep soil temperatures are set from an input field as outlined in the experiments

below.

## 2.2 AMIP simulations

All experiments undertaken in this study are summarised in Table 1 for ease of reference. More details on these simulations are given in Sections 2.2.1, 2.2.3, 2.2.4 and 2.2.5, below.

### 2.2.1 "Free land" simulations

The following simulations are undertaken with freely varying land conditions ("land conditions" refers to surface temperature, soil temperature and soil moisture from here on), i.e. Eqs. 1–4 are used by the model.

(1) AMIP run: An AMIP run using prescribed, observational SSTs and sea ice concentrations from 1979 to 2008 (30 years long). $CO_2$ concentrations are set to 346 ppmv and, sulphur dioxide, soot and biomass burning aerosol emissions are representative of those for the year 2000 C.E. Land conditions are allowed to vary freely. The experiment is denoted as

**A** from here on.





**Table 1.** A summary of the experimental specifications. In the sea surface temperature (SST) column, A refers to SSTs from the AMIP run and A4K to those of the AMIP+4K (A4K) run. 'FREE' refers to freely varying land temperatures and soil moisture. Plant physiology is set to 'ON' when vegetation responds to $CO_2$ changes and 'OFF' when it uses the default value (346 ppmv) i.e. only atmospheric radiation responds to higher $CO_2$. Experiments are ordered following the descriptions in Sections 2.2.1, 2.2.3, 2.2.4 and 2.2.5.

| Run I.D. [run length: years] | SST | Land Conditions | $CO_2$ [ppmv] | Plant Physiology | Solar Constant [W m$^{-2}$] |
|---|---|---|---|---|---|
| Free land simulations (Section 2.2.1). | | | | | |
| A [30] | A | FREE | 346 | ON | 1365 |
| A4K [30] | A4K (i.e. AMIP+4K) | FREE | 346 | ON | 1365 |
| A4x [30] | A | FREE | 1384 | ON | 1365 |
| Arad4x [30] | A | FREE | 1384 | OFF | 1365 |
| Asc [30] | A | FREE | 346 | ON | 1410.7 |
| Prescribed land simulations (Section 2.2.3). | | | | | |
| $A_{PL}$ [29] | A | A | 346 | ON | 1365 |
| $A4K_{PL4K}$ [29] | A4K | A4K | 346 | ON | 1365 |
| $A4x_{PL4x}$ [29] | A | A4x | 1384 | ON | 1365 |
| $Arad4x_{PLrad4x}$ [29] | A | Arad4x | 1384 | OFF | 1365 |
| $Asc_{PLsc}$ [29] | A | Asc | 346 | ON | 1410.7 |
| Single forcing experiments (Section 2.2.4). | | | | | |
| $A4K_{PL}$ [29] | A4K | A | 346 | ON | 1365 |
| $A_{PL4K}$ [29] | A | A4K | 346 | ON | 1365 |
| $A4x_{PL}$ [29] | A | A | 1384 | ON | 1365 |
| $A_{PL4x}$ [29] | A | A4x | 346 | ON | 1365 |
| $Arad4x_{PL}$ [29] | A | A | 1384 | OFF | 1365 |
| $A_{PLrad4x}$ [29] | A | Arad4x | 346 | OFF | 1365 |
| $Asc_{PL}$ [29] | A | A | 346 | ON | 1410.7 |
| $A_{PLsc}$ [29] | A | Asc | 346 | ON | 1365 |
| Uniform surface temperature experiments (Section 2.2.5). | | | | | |
| $A4K_{PLU4K}$ [29] | A4K | A+4K | 346 | ON | 1365 |
| $A_{PLU4K}$ [29] | A | A+4K | 346 | ON | 1365 |



(2) AMIP4K run: The same as A but a uniform 4 K added to the SST field (denoted as **A4K** from here on).

(3) AMIP4xCO$_2$ run: The same as A but CO$_2$ is quadrupled to 1384 ppmv (denoted as **A4x** from here on).

(4) AMIP4xCO$_2$ no plant physiological response i.e. radiative (rad) only: The same as A4x but the plant physiological response to CO$_2$ is switched off (as described in Andrews et al., 2011; Boucher et al., 2009; Doutriaux-Boucher et al.,
5        2009, and denoted as **Arad4x** from here on).

(5) AMIP +3.3% solar constant: The same as A except the solar constant is increased by $\sim$3.3% to 1410.7 W m$^{-2}$ as done by Andrews et al. (2012b), which gives a similar sized radiative forcing to the 4xCO$_2$ experiments (denoted as **Asc** from here on).

All AMIP simulations were initialised with conditions from 1$^{st}$ October 1978 and run until the end of December 2008.

## 2.2.2    Specifications for generating the prescribed land conditions

In order to generate the necessary fields to prescribe the land conditions, instantaneous values of the surface temperature on each tile and, soil temperature and moisture (on each soil level) are output every three hours from experiments (1)–(5) above. In the "prescribed land" simulations, the land conditions are read in by the model every 3 hours and updated (by interpolation) every hour (two time steps). Furthermore, land conditions from the first 15 months of the AMIP free land simulations are not
used (i.e. the prescribed land simualtions are run from January 1980 to December 2008, inclusive) to ensure that no impacts from the land scheme "spinning up" are included in the prescribed runs. The surface temperature, soil moisture and soil temperatures are all prescribed every 3 hours for the whole period 1980–2008 to minimise the differences between free and prescribed land simulations. The interpolated, 3-hourly data are used instead of time step (30 minute) data due to limitations of reading in such large datasets in the current ACCESS1.0 framework. The prescribed land conditions experiments will
therefore not be identical to the free land simulations. Nevertheless, earlier work by Ackerley and Dommenget (2016) note that a simulation with temperatures updated each time step is "almost climatologically indistinguishable" from another using 3-hourly data. Therefore, corresponding free and prescribed land simulations should be climatologically alike, which is evaluated in Section 3. Finally, land surface temperatures are not prescribed over permanent ice sheets (Antarctica and Greenland) to avoid the development of negative temperature biases that are discussed in Ackerley and Dommenget (2016). The impact of
not specifying the land temperatures under the ice sheets is likely to be negligible and is discussed in Section 3. The input data fields are available by following the instructions in the *Code and data availability* Section.

### 2.2.3    AMIP prescribed land simulations

All simulations that have prescribed land conditions are denoted with a "PL". The AMIP prescribed land simulations use Eq. 5 insetad of Eq. 1, both $\Delta T_{*EVAP}$ and $\Delta T_{*MLT}$ set to zero and, the following boundary conditions are used:

(6) AMIP prescribed land run: The same as A except land conditions are also prescribed from A. Experiment is denoted as **A$_{PL}$** from now.





(7) AMIP4K prescribed land run: As A4K except land conditions are prescribed using the output from A4K. Experiment denoted as **A4K$_{PL4K}$** from now.

(8) AMIP4xCO$_2$ prescribed land run: As A4x except land conditions are prescribed using the output from A4x. Experiment is denoted as **A4x$_{PL4x}$** from now.

(9) AMIP4xCO$_2$ no plant physiological response prescribed land run: As Arad4x except land conditions are prescribed using the output from Arad4x. Experiment is denoted as **Arad4x$_{PLrad4x}$** from now.

(10) AMIP +3.3% solar constant prescribed land run: As Asc except land conditions are prescribed using the output from Asc. Experiment is denoted as **Asc$_{PLsc}$** from now.

### 2.2.4 Combinations of AMIP land and ocean conditions ("combined" experiments)

In these experiments, different combinations of land, SST, atmospheric CO$_2$ and solar irradiance boundary conditions are used. These experiments were designed to single out the impact of the land response to a forcing on the atmosphere or the impact of that forcing agent without the land responding. Again (as in Section 2.2.3), Eq. 5 instead of Eq. 1 and, both $\Delta T_{*EVAP}$ and $\Delta T_{*MLT}$ are set to zero for these simulations. The boundary condidions used in these experiments are:

(11) SST field from A4K and land conditions from A. From now, denoted as **A4K$_{PL}$**.

(12) SST field from A and land conditions from A4K. From now, denoted as **A$_{PL4K}$**.

(13) SST and land conditions from A with CO$_2$ concentrations the same as in A4x. From now, denoted as **A4x$_{PL}$**.

(14) SST and CO$_2$ concentrations the same as A and land conditions from A4x. From now, denoted as **A$_{PL4x}$**.

(15) SST and CO$_2$ concentrations (no plant response) from Arad4x and land conditions from A. From now, denoted as **Arad4x$_{PL}$**.

(16) SST and CO$_2$ concentrations the same as A and land conditions from Arad4x. From now, denoted as **A$_{PLrad4x}$**.

(17) SST and land conditions from A and solar constant as in Asc. From now, denoted as **Asc$_{PL}$**.

(18) SST and land conditions from Asc and solar constant as in A. From now, denoted as **A$_{PLsc}$**.

### 2.2.5 Uniform surface temperature perturbation ("uniform" experiments)

An extra two experiments are undertaken to identify the impact of applying a uniform increase in temperature over the land
only (analogous to the AMIP4K SST experiment but for the land) and a uniform global increase in surface temperature (i.e. global warming with minimal land-sea contrast). As in Sections 2.2.3 and 2.2.4, Eq. 5 is used instead of Eq. 1 and, both $\Delta T_{*EVAP}$ and $\Delta T_{*MLT}$ set to zero for these simulations. The boundary conditions used in these experiments are:



(19) Uniform increase in land surface temperatures from A by 4 K and SST field from A4K. From now, denoted as **A4K$_{\mathbf{PLU4K}}$**.

(20) Uniform increase in land surface temperatures from A by 4 K and SST field from A. From now, denoted as **A$_{\mathbf{PLU4K}}$**.

In both experiments (19) and (20), soil temperatures and moisture are prescribed from the A experiment.

## 2.3 Reference datasets

ERA-Interim data are taken from 1980–2008 (Dee et al., 2011) for both the surface air temperature (TAS) and pressure at mean sea level (PSL) for comparison with the A and A$_{PL}$ simulations. ERA-Interim reanalysis data have been used to evaluate TAS globally for the $5^{th}$ Coupled Model Intercomparison Project (Flato et al., 2013). ERA-Interim data provide a globally complete (unlike surface observations which are heterogeneously spread), observationally constrained (as is PSL) dataset for comparison with the simulations in this study. Furthermore, there is good agreement between reanalysis-derived TAS and gridded data from

station-based estimates (Simmons et al., 2010), which suggests the ERA-Interim derived TAS is a reliable dataset.

For precipitation, the Climate Prediction Centre Merged Analysis of Precipitation (CPC CMAP Xie and Arkin, 1997; Arkin et al., 2018) data, for the years 1980–2008 inclusive, are used. The CMAP data are derived from a combination of satellite-based instruments. It is important to note that, while there are biases in any reference dataset and others could be used (e.g. GPCP or CMORPH for rainfall, see Adler et al., 2003; Joyce et al., 2004, respectively), the focus of the paper is not to

explore the model biases themselves. The reference datasets are simply used to show that there is no negative impact on the simulated climate (relative to the free land simulations) when the land conditions are prescribed.

## 3 Verification of the AMIP prescribed land runs

### 3.1 Surface air temperature: TAS

The difference (A-ERA-Interim) in grid-point mean (averaged over all simulated years) TAS is plotted in Fig. 1. Positive

anomalies ($\sim$0.5 K) are visible over many ocean basins but the largest differences are over the land ($>$ 1 K magnitude over North Africa, Antarctica and the Himalaya). Nevertheless, the temperature biases in Fig. 1(a) are consistent with those presented in Flato et al. (2013) from the CMIP5 multi-model mean (their Fig. 9.2(b)) and the global mean RMSD of 1.68 K (Table 2) is also comparable to the mean absolute grid-point errors of 1–3 K also given in Flato et al. (2013) (their Fig. 9.2(c)). The largest model errors primarily occur in the regions that have the largest uncertainties in the ERA-Interim TAS dataset (e.g.

North Africa, Antarctica and the Himalaya— Flato et al., 2013, their Fig. 9.2(d)). Finally, the pattern correlation between A and ERA-Interim fields is approximately 1 (Table 2), which indicates that relatively low and high surface temperatures are simulated in the correct geographical locations. Overall therefore, the TAS field in the ACCESS1.0 AMIP simulation (and the biases) are consistent with those of other models.



The difference in TAS for $A_{PL}$ relative to A is ploted in Fig. 1(b). It is immediately obvious that the differences in TAS[1]
between $A_{PL}$ and A are much smaller than those between A and ERA-Interim (Fig. 1(a)). There are also very few places where
the differences are statistically significant in Fig. 1(b). Furthermore, the RMSD is much larger between A and ERA-Interim
than between $A_{PL}$ and A (1.69 K and 0.13 K, respectively in Table 2). Overall, in terms of TAS, the A and $A_{PL}$ simulations

are climatologically very similar such that the inter-model differences are much smaller than the model-reanalysis differences.

Each of the "prescribed land" (PL) simulations (A4K$_{PL4K}$, A4x$_{PL4x}$, Arad4x$_{PLrad4x}$ and Asc$_{sc}$, described in Section 2.2.3)
are compared with their corresponding free land simulations (A4K, A4x, Arad4x and Asc, respectively, Section 2.2.1) in order
to validate them. The differences in TAS are non-significant over the vast majority of the globe for the prescribed versus free
land simulations (Figs. 1(c)–(f)). Moreover, the RMSD between each experiment pair is 0.11 K with pattern correlations of

unity or close to unity (see Table 2). Therefore, the values of TAS in the A4K$_{PL4K}$, A4x$_{PL4x}$, Arad4x$_{PLrad4x}$ and Asc$_{PLsc}$
runs are almost climatologically indistinguishable from those of A4K, A4x, Arad4x and Asc, respectively (as intended).

In order to further validate whether the PL simulations adequately reproduce the climate of their free land counterparts under
different boundary conditions (i.e. SST+4K, 4xCO$_2$ and +3.3% insolation), the differences in TAS between corresponding free
and prescribed land pairs (e.g. [A4K$_{PL4K}$-$A_{PL}$]-[A4K-A]) are plotted in Figs. 2(a)–(d). Furthermore, the RMSD and pattern

correlations for the differences in TAS between those corresponding prescribed and free land pairs are given in Table 3. Pattern
correlations are proximately 1 for all experiment pairs (Table 3). Furthermore, the RMSD values are <0.1 K, which is a similar
magnitude to the differences plotted in Fig. 1(c)–(f) and smaller than the differences in TAS associated with each change in
boundary condition (see Figs. S1(a)–(d) and S2(a)–(d), Supplementary Material). Therefore, the changes in TAS for A4K$_{PL4K}$,
A4x$_{PL4x}$, Arad4x$_{PLrad4x}$ and Asc$_{PLsc}$ relative to $A_{PL}$ are almost identical to those of A4K, A4x, Arad4x and Asc relative to

A (compare Figs. S1(a)–(d) and S2(a)–(d), Supplementary Material). Overall, the responses of TAS to the perturbed SST, CO$_2$
and insolation in the prescribed land simulations are very similar to those in the free land simulations.

## 3.2   Precipitation: PR

### 3.2.1   Prescribed vs free land experiment pairs

Differences between the A simulation and CMAP precipitation fields are plotted in Fig. 1(g). Precipitation is too high over the

western Indian Ocean, the northern Tropical Pacific and within the mid-latitudes of both hemispheres. Conversely, precipitation
is too low over the south-western Maritime Continent, central Africa, Amazonia and over the Antarctic. The precipitation biases
over the western Indian Ocean and Amazonia are also visible in the CMIP5 multi-model mean (see Fig. 9.4(b) in Flato et al.,
2013). The rainfall biases in the remaining regions (listed above) are consistent with those presented in Walters et al. (2011) for
HadGEM2-A (the model from which ACCESS1.0 is derived, see Bi et al., 2013). The RMSD is 1.25 mm day$^{-1}$ (Table 2) for

A relative to CMAP, which is consistent with the values presented for HadGEM2-A by Walters et al. (2011) (2.02 mm day$^{-1}$

---

[1]Note: the calculation of TAS is performed by interpolating between the surface temperature and that of the lowest model level in ACCESS1.0, therefore
changes in the temperature at level 1 may also change TAS even if surface temperatures are unchanged.



**Table 2.** The area-weighted root-mean-squared-differences (RMSD) and pattern correlations (PC) for surface air temperature (TAS), precipitation (PR) and mean sea level presure (PSL) for the A and $A_{PL}$ simulations relative to the "obervational" (OBS) reference datasets (rows 2 and 3). Rows 4–8: the RMSDs and PCs for each "prescribed land" simulation relaitive to its counterpart "free land" simulation (experiment names defined in Section 2.

| Difference between | RMSD TAS (K) | PC TAS | RMSD PR (mm day$^{-1}$) | PC PR | RMSD PSL (hPa) | PC PSL |
|---|---|---|---|---|---|---|
| A - OBS | 1.68 | $\approx 1$ | 1.25 | 0.92 | 2.40 | $\approx 1$ |
| $A_{PL}$ - OBS | 1.69 | $\approx 1$ | 1.26 | 0.92 | 2.48 | $\approx 1$ |
| $A_{PL}$ - A | 0.13 | 1.00 | 0.28 | $\approx 1$ | 0.45 | 1.00 |
| A4K$_{PL4K}$ - A4K | 0.11 | $\approx 1$ | 0.27 | $\approx 1$ | 0.31 | $\approx 1$ |
| A4x$_{PL4x}$ - A4x | 0.11 | 1.00 | 0.30 | 0.99 | 0.44 | 1.00 |
| Arad4x$_{PLrad4x}$ - Arad4x | 0.11 | $\approx 1$ | 0.27 | $\approx 1$ | 0.31 | $\approx 1$ |
| Asc$_{PLsc}$ - Asc | 0.11 | 1.00 | 0.28 | $\approx 1$ | 0.47 | 1.00 |

"$\approx 1$" implies that the correlation coefficient is not unity, but rounds to unity when only two decimal places are considered.

for JJA and 1.54 mm day$^{-1}$ for DJF, relative to GPCP data). Overall, the precipitation biases in the A simulation are consistent with those in other GCMs.

The differences in precipitation between $A_{PL}$ and A are plotted in Fig. 1(h) and (as with TAS) it is clear that almost none of the differences in precipitation are significant. Furthermore, the RMSD between $A_{PL}$ and CMAP is almost identical to that
of A relative to CMAP and, the RMSD for $A_{PL}$ relative to A is smaller by almost a factor of five (see Table 2) than relative to CMAP. The pattern correlations between $A_{PL}$ and A are also approximately equal to one, which shows that regions with relatively high and low precipitation (climatologically) are almost identical in the two respective simulations. Therefore, the differences in PR between $A_{PL}$ and A are small in terms of the climatological mean.

As with TAS, the differences in PR between other prescribed land simulations (A4K$_{PL4K}$, A4x$_{PL4x}$, Arad4x$_{PLrad4x}$ and
Asc$_{sc}$) and their respective free land runs (4K$_{PL4K}$, A4x$_{PL4x}$, Arad4x$_{PLrad4x}$ and Asc$_{sc}$) are plotted in Figs. 1(i)–(l). Very few of the differences in PR are statistically significant; however, there is an increase in precipitation over Amazonia in all of the prescribed land runs relative to their free land counterparts. A similar region of higher precipitation over Amazonia between prescribed and free land simulations is also seen in Ackerley and Dommenget (2016). Given that there is no change in surface temperature or soil moisture (both prescribed) it may be that rainwater is accumulating in the vegetation canopy
and being re-evaporated (see Cox et al., 1999). Indeed, there is an increase in the latent heat flux over the region with higher precipitation in all of the prescribed land simulations relative to the free land simulations (see Fig. S3, Supplementary Material). This is a systematic bias in the prescribed land simulations relative to their free land counterparts; however, the precipitation is approximately 1–2 mm day$^{-1}$ higher in the prescribed land runs, which almost exactly offsets the $\sim$2 mm day$^{-1}$ bias for the A simulation relative to CMAP (Fig. 1(g)). Therefore, the prescribed land simulation is closer to the observed estimate
than the free land simulation. A more detailed investigation into Amazonian rainfall is beyond the scope of this current general



overview and evaluation paper, but such a study may be useful to understand the dry bias over the Amazon in the free land
simulations.

As with TAS, the RMSD and pattern correlations for the differences in PR between corresponding prescribed and free land
pairs (e.g. [A4K$_{PL4K}$-A$_{PL}$]-[A4K-A]) are given in Table 3. The pattern correlations lie between 0.8 and 0.95 (Table 3) for

the change in PR between the perturbed PL simulations (A4K$_{PL4K}$, A4x$_{PL4x}$, Arad4x$_{PLrad4x}$ and Asc$_{PLsc}$) and their free
land counterparts (A4K, A4x, Arad4x and Asc), relative to their respecitve control simulations (A$_{PL}$ and A). Furthermore, the
RMSD values lie in the range 0.22 – 0.38 mm day$^{-1}$, which is a similar magnitude to the differences plotted in Fig. 1(c)–(f)
and Figs. 2(e)–(h). Therefore, the differences between corresponding prescribed and free land simulations (e.g. A4K$_{PL4K}$ and
A4K) are much smaller than the PR differences caused by the boundary condition changes (see Figs. S1(e)–(h) and S2(e)–(h),

Supplementary Material). The lower pattern correlation values and higher RMSDs for PR relative to TAS are likely to be due
to TAS being more highly constrained by the prescribed surface temperatures than PR (i.e. TAS is diagnostically calculated
from the surface temperature and the temperature of the lowest model level).

For further verification, the changes in global, ocean and land mean precipitation are presented in Table 4. The differences in
precipitation between the free land and PL experiment pairs are all the same sign (i.e. corresponding positive or negative) and

lie within $\pm 0.08$ mm day$^{-1}$ (i.e. small). The largest difference occurs over land in A4K$_{PL4K}$ experiment where the increase
in precipitation (relative to A$_{PL}$) is statistically significant whereas, for A4K relative to A, it is not. The higher precipitation
over the Amazon (Fig. 2(e)) is likely to be contributing to the higher land-mean precipitation in A4K$_{PL4K}$ relative to A4K.
Conversely, the dry bias over the Amazon in the free land simulations may equally be a factor for the muted response of the
mean precipitation over land in the A4K experiment relative to A4K$_{PL4K}$. Again, a more detailed investigation into Amazonian

rainfall biases is beyond the scope of this study; however, given the sensitivity of this region to model configuration and climate
change (see Good et al., 2013) the prescribed land simulation may be a useful tool to investigate Amazon precipitation further.
Another point of note is that precipitation increases significantly in the runs without plant physiological responses to CO$_2$ but
does not change in those without (Table 4). In the A4x and A4x$_{PL}$ experiments, plant stomata respond to increasing CO$_2$
by narrowing and thereby reducing moisture availability for precipitation from transpiration. In Arad4x and Arad4x$_{PLrad4x}$

however, the stomatal response is switched off and so evapotranspiration can increase in response to land surface warming,
as can precipitation. These results are consistent with those of Doutriaux-Boucher et al. (2009), Boucher et al. (2009) and
Andrews et al. (2011).

### 3.3 Pressure at mean sea level: PSL

#### 3.3.1 Prescribed vs free land experiment pairs

The difference in PSL for A relative to ERA-Interim is plotted in Fig. 1(m) in order to provide a surface-based indication of
changes in the atmospheric circulation (as also done in Collins et al., 2013). The RMSD for A relative to ERA-Interim is 2.4
hPa; however, the pattern correlation is almost unity (see Table 2) and indicates that regions with relatively high and low PSL
correspond well. There are several biases in the PSL field, nonetheless. Positive PSL anomalies are visible in A relative to ERA-



**Table 3.** The area-weighted root-mean-squared-differences (RMSD) and pattern correlations (PC) for the response in the climate to the perturbed boundary conditions (SST+4K, 4xCO$_2$ and +3.3% solar constant (Section 2) for each "prescribed land" pair relative to the corresponding "free land" pair.

| Difference between | RMSD TAS (K) | PC TAS | RMSD PR (mm day$^{-1}$) | PC PR | RMSD PSL (hPa) | PC PSL |
|---|---|---|---|---|---|---|
| (A4K$_{PL4K}$-A$_{PL}$) - (A4K-A) | 0.08 | $\approx$1 | 0.38 | 0.92 | 0.45 | 0.96 |
| (A4x$_{PL4x}$-A$_{PL}$) - (A4x-A) | 0.09 | $\approx$1 | 0.27 | 0.89 | 0.38 | 0.92 |
| (Arad4x$_{PLrad4x}$-A$_{PL}$) - (Arad4x-A) | 0.08 | 0.99 | 0.22 | 0.88 | 0.35 | 0.91 |
| (Asc$_{PLsc}$-A$_{PL}$) - (Asc-A) | 0.08 | 0.99 | 0.25 | 0.83 | 0.33 | 0.91 |

"$\approx$1" implies that the correlation coefficient is not unity, but rounds to unity when only two decimal places are considered.

Interim over the Arctic (largest anomaly around 90°E), the north Pacific, northern Africa and the Mediterranean and, between 30°S–60°S in each ocean basin (see Fig. 1(m)). There are negative anomalies over central and southern Africa, South America, North America and Antarctica. The PSL anomalies though, are consistent with those presented in Martin et al. (2006) (their Fig. 6), who used a higher-resolution (half the grid spacing of ACCESS1.0) version of HadGEM2 (from which ACCESS1.0 is
developed—see Bi et al., 2013).

The RMSD (2.48 hPa) and pattern correlations ($\approx$1) for the A$_{PL}$ simulation are almost identical to those of A relative to ERA-Interim. Furthermore, the RMSD between A$_{PL}$ and A is 0.45 hPa and the pattern correlation is unity (Table 2), which indicates that the PSL field is reproduced well in the A$_{PL}$ simulation relative to A. The main difference in the PSL fields between A$_{PL}$ and A occurs over the Arctic (Fig. 1(n)), which is consistent with the lower temperatures there (see Fig. 1(b)).
Nevertheless, over the vast majority of the globe, the differences in the simulated PSL field between A$_{PL}$ and A are not statistically significant.

The RMSDs for each of the other corresponding PL and free-land simulations (e.g. A4K$_{PL4K}$ versus A4K) lie between 0.3–0.5 hPa with pattern correlations of close to unity (see Table 2). The magnitudes and distribution of PSL in the PL simulations therefore compare well with their free land counterparts (as with A$_{PL}$ versus A). In terms of grid-point PSL values, the largest
differences occur in the northern and southern polar regions (see Figs. 1(o)–(r)); however, the differences in PSL are not statistically significant over the vast majority of grid points. Overall, the small differences in the PSL fields between the PL and free land simulations suggest that the simulated, climatological global circulations are very similar.

Again (as with TAS and pr), the RMSD and pattern correlations for the differences in PSL between corresponding prescribed and free land pairs (e.g. [A4K$_{PL4K}$-A$_{PL}$]-[A4K-A]) are given in Table 3. The RMSD between the change in PSL associated
with each boundary condition perturbation for the PL simulations relative to their free land counterparts lie between 0.33 and 0.45 hPa (Table 3). The largest RMSD for PSL changes (0.45 hPa) occurs in the SST+4K experiments (i.e. [A4K$_{PL4K}$-A$_{PL}$] relative to [A4K-A]); however, the changes in PSL associated with increasing global SSTs are much larger (approximately $\pm$3.5 hPa, see Fig. S2(i), Supplementary Material) than the RMSD. The changes in PSL associated with quadrupling CO$_2$ are $\pm$2.5 hPa (Figs. S2(j) and (k)) are larger than the RMSD between the corresponding prescribed and free land simulations (0.38
hPa and 0.35 hPa, see Table 3). The smallest changes in PSL occur in the increased solar constant simulations (around $\pm$1.5



**Table 4.** The difference in global, land points and sea points mean precipitation, mm day$^{-1}$ [%] for each of the specified simulations in rows 1, 5, 9 and 13 (details of each simulation are given in Section 2). Numbers in italics and marked with an asterisk are not statistically significant using the Student's t-test (p>0.05).

| Region | A4K-A | A4K$_{PL4K}$-A$_{PL}$ |
|---|---|---|
| Global mean | 0.38 [12.33] | 0.38 [12.32] |
| Land mean | *0.01 [0.33]\** | 0.09 [4.04] |
| Sea mean | 0.53 [15.16] | 0.50 [14.37] |
| Region | A4x-A | A4x$_{PL4x}$-A$_{PL}$ |
| Global mean | -0.19 [-6.11] | -0.18 [-5.94] |
| Land mean | *0.00 [-0.14]\** | *0.02 [0.86]\** |
| Sea mean | -0.27 [-7.52] | -0.27 [-7.63] |
| Region | Arad4x-A | Arad4x$_{PLrad4x}$-A$_{PL}$ |
| Global mean | -0.13 [-4.31] | -0.14 [-4.40] |
| Land mean | 0.10 [4.80] | 0.11 [4.97] |
| Sea mean | -0.23 [-6.47] | -0.24 [-6.72] |
| Region | Asc-A | Asc$_{PLsc}$-A$_{PL}$ |
| Global mean | -0.05 [-1.61] | -0.05 [-1.66] |
| Land mean | 0.15 [7.58] | 0.16 [7.51] |
| Sea mean | -0.13 [-3.78] | -0.14 [-3.93] |

hPa, Fig. S2(l)) and likewise, the lowest RMSD between the PL and free land simulations (0.33 hPa, see Table 3). Finally, the pattern correlations between the PL and free land simulations are all >0.9 (column 7, Table 3), which shows that the spatial changes in PSL associated with each boundary condition change are also very similar. The largest grid-point differences in PSL primarily occur in polar regions, where surface temperatures are not prescribed (Figs. 2(i)–(l)); however, the differences

5 in PSL are not statistically significant over the majority of the globe.

### 3.4 Vertical profiles: Global, ocean-only and land-only means

As a final validation, the vertical changes in mean air temperature (ta) associated with the SST+4K, 4xCO$_2$, 4xCO$_2$rad and +3.3% insolation are plotted for the PL (red lines) and free land (black lines) in Fig. 3. Furthermore, the ta profile differences are compared with results from other studies (where available) for further validation of these simulations.

10 The global, ocean and land mean changes in ta for A4K-A are almost identical to those of A4K$_{PL4K}$-A$_{PL}$ (values lie within approximately ±0.1 K, see Figs. 3(a)–(c)). Furthermore, ta values are higher at all levels from 1000 hPa to 200 hPa, with the largest increase around 300 hPa. Overall, atmospheric dry stability increases as a result of increasing global SST by 4 K



both globally and over the ocean with a slight decrease in dry stability over land between approximately 1000 to 500 hPa. The changes to the ta profiles in both the PL (A4K$_{PL4K}$-A$_{PL}$) and free simulations (A4K-A) agree with those described in Dong et al. (2009) and He and Soden (2015).

The differences in ta between the prescribed (red lines) and free (black lines) land for the 4xCO$_2$ experiments (both with and

without plant physiology) are plotted in Figs. 3(d)–(i). As with the SST+4K experiments, the differences between the prescribed and free land simulations are small ($\sim \pm 0.1$ K) and primarily restricted to the land in the A4x$_{PL4x}$ and A4x experiments. The largest changes in ta from quadrupling atmospheric CO$_2$ occur around 850 hPa for the global and ocean mean regardless of whether the plant physiological response to CO$_2$ is included or not (Figs. 3(d), (e), (g) and (h)) in agreement with Dong et al. (2009), Kamae and Wanatabe (2013), Richardson et al. (2016) and Tian et al. (2017).

Finally, the ta profiles for the 3.3% increase in insolation simulations (Asc and Asc$_{PLsc}$ relative to A and A$_{PL}$, respectively) are plotted in Figs. 3(j)–(l). Again, the differences between the free and prescribed land simulations are small ($\sim \pm 0.1$ K) and the vertical distribution of ta changes are almost identical. Atmospheric dry stability increases globally and over the ocean, with the largest increases in ta around 300 hPa (Figs. 3(j) and (k)), which compares well with the model results of Cao et al. (2012). Conversely, air temperatures increase uniformly by approximately 0.8 K from 950 – 500 hPa in both the Asc and Asc$_{PLsc}$

simulations (Fig. 3(l)) over the land; however, dry static stability increases around 300 hPa (again in agreement with Cao et al., 2012).

Overall, the differences in ta between the prescribed and free land simulations are small relative to the changes associated with each boundary condition change. Furthermore, the changes in ta in both the prescribed and free land simulations are consistent with those in other studies.

## 4   Surface air temperature changes in the "combined" and "uniform" experiments

Only the changes in surface air temperature are discussed below for each of the "combined" and "uniform" temperature perturbation experiments (outlined in Sections 2.2.4 and 2.2.5, respectively) to verify that the temperature repsonse is consistent with the imposed boundary conditions. The changes in precipitation and circulation associated with these experiments are to be discussed in a future piece of work (Chadwick et al., in prep.).

### 4.1   "Combined" experiments

Changes in TAS over the land can be seen in the experiments that use land conditions from the AMIP runs with changed boundary conditions i.e. A4K$_{PL}$, A4x$_{PL}$, Arad4x$_{PL}$ and Asc$_{PL}$ (Figs. 4(a)–(d)). As the calculation of TAS is performed by interpolating between the surface temperature and that of the lowest model level in ACCESS1.0, changes in the temperature at level 1 will change TAS even if surface temperatures are unchanged. This explains why TAS increases over the land in A4K$_{PL}$,

as the global atmosphere will warm from increased SST (Fig 4(a)). There are also positive TAS anomalies over high-latitudes in all the experiments plotted in Fig. 4 relative to A$_{PL}$, which is unsurprising as the snow cover and surface temperatures are not prescribed there. The changes in TAS are also higher over the ocean than the land (land/sea contrast is 0.25).





The changes in TAS for A4x$_{PL}$, Arad4x$_{PL}$ and Asc$_{PL}$ are not statistically significant over the majority of the land surface and may be related to adjustments in the surface sensible and latent heat fluxes as the atmosphere responds to the increase in $CO_2$ concentrations or insolation (Figs. 4(b)–(d)). Conversely, the changes in TAS over the land are statistically significant and positive in all runs with perturbed land surface conditions (Figs. 4(e)–(h)). Overall, relative to A$_{PL}$, the changes in TAS

for the simulations described in Section 2.2.4 (plotted in Fig. 4) are consistent with the land surface and boundary condition perturbations imparted upon them.

### 4.2   "Uniform" experiments

The spatial differences in TAS are plotted in Fig. 5(a) for the A4K$_{PLU4K}$ simulation relative to A$_{PL}$. The changes in TAS over the land and the sea are very similar with a land-sea thermal contrast of 0.9. The main difference in TAS between the land

and the ocean is over Antarctica and Greenland where the surface temperatures not prescribed and the temperature change is muted.

In the A$_{PLU4K}$ experiment (relative to A$_{PL}$), TAS increases over all land points by 1.5 – 4.5 K (statistically significant) except over Antarctica and Greenland where temperatures are not prescribed (Fig. 5(b)). Another interesting feature of this simulation is that the land-sea thermal contrast is very large (with a value of 40); however, the large contrast is unsurprising

given the large temperature increase is only applied to the land.

### 5   Summary, conclusions and future work

This paper has outlined the results of a novel set of AMIP-type model simulations that use prescribed SSTs and land surface fields (surface temperature, soil temperature and soil moisture). The main results of this study are:

(1)   The differences in climate between the simulations with freely varying land conditions and their prescribed land coun-
terparts (e.g. A vs A$_{PL}$) are much smaller than the underlying systematic errors relative to the observational datasets (i.e. A vs OBS). Therefore, prescribing the land conditons does not degrade the model-simulated climate.

(2)   The changes in global mean precipitation and vertical temperature profiles in the A4K, A4x, Arad4x and Asc experiments are almost identical to those of their corresponding prescribed land simulations—A4K$_{PL4K}$, A4x$_{PL4x}$, Arad4x$_{PLrad4x}$ and Asc$_{PLsc}$.

(3)   The changes in TAS associated with holding the land fixed while changing a forcing agent (e.g. A4x$_{PL}$) or fixing the forcing agent and using the land response to that agent (e.g. A$_{PL4x}$) are consistent with imposed state and are therefore applied correctly.

(4)   The "U4K" experiments (results described in Section 4.2) provide a novel extension to the A4K experiment where the land-sea thermal contrast is suppressed; however, the TAS response is very similar to that of the A4K$_{PL4K}$ experiment.



(5) Likewise, the $A_{PLU4K}$ simulation resembles the TAS response in the $Asc_{PLsc}$ experiment, except the magnitude of the climatic changes are larger in $A_{PLU4K}$.

Overall, this study has presented a set of experiments that could be used to answer questions about the separate roles of the land, ocean and atmosphere under climate change. While this study evaluates those simulations, it does not provide an in-depth
scientific analysis of all the model simulations undertaken. By providing those data for others to download, it is the intention of this paper to provide a background analysis for validation purposes and to provide information on how to acquire these data. These simulations may also help to answer some of the key questions arising from the CFMIP and CMIP initiatives (see Eyring et al., 2016; Webb et al., 2017, respectively) given in Section 1 and to provide a better understanding of the regional drivers of precipitation over the land.

*Code and data availability.*   The model source code for ACCESS is not publicly available; however, more information can be found through the ACCESS-wiki at https://accessdev.nci.org.au/trac/wiki/access. Any registered ACCESS users who wish to gain access to the source code described in this paper can do so from the following:

**For A, A4K and A4x**

$https://access-svn.nci.org.au/svn/um/branches/dev/cxf565/r3909\_my\_vn7.3@4793$

**For Arad4x**

$https://access-svn.nci.org.au/svn/um/branches/dev/dxa565/src\_plant\_co2/src@10276$

**For Asc**

$https://access-svn.nci.org.au/svn/um/branches/dev/dxa565/src\_solcnst/src@10274$

**For $A_{PL}$, $A4K_{PL4K}$, $A4x_{PL4x}$, $A4K_{PL}$, $A_{PL4K}$, $A4x_{PL}$, $A_{PL4x}$, $A_{PLrad4x}$, $A_{PLsc}$, $A4K_{PLU4K}$ and $A_{PLU4K}$**

$https://access-svn.nci.org.au/svn/um/branches/dev/dxa565/src\_presT\_reg/src@9826$

**For $Arad4x_{PLrad4x}$ and $Arad4x_{PL}$**

$https://access-svn.nci.org.au/svn/um/branches/dev/dxa565/src\_presT\_reg\_np/src@10269$

**For $Asc_{PLsc}$ and $Asc_{PL}$**

$https://access-svn.nci.org.au/svn/um/branches/dev/dxa565/src\_presT\_reg\_sc/src@10272$

Data are publicly available from the National Computational Infrastructure (NCI) (see Ackerley, 2017). Input surface temperature, soil moisture and deep soil temperatures are also available from the NCI upon request (also refer to Ackerley, 2017). The relevant doi (and other metadata) for each of the individual experiments can be found in the supplementary file attached to this paper ($plamip\_expts\_doi\_list.xlsx$). Use of these data in any publications requires both a citation to this article and an appropriate acknowledgement to the data resource page (see Ackerley, 2017, for more details on acknowledging the dataset)

*Competing interests.*   The authors declare that they have no conflict of interest.





*Acknowledgements.* This project was primarily funded by the ARC Centre of Excellence for Climate System Science (CE110001028). Duncan Ackerley also acknowledges the Met Office for funding time to complete this work through the Joint BEIS/Defra Met Office Hadley Centre Climate Programme (GA01101). The ACCESS simulations were undertaken with the assistance of the resources from the National Computational Infrastructure (NCI), which is supported by the Australian Government. Robin Chadwick was supported by the Newton Fund through the Met Office Climate Science for Service Partnership Brazil (CSSP Brazil). We would also like to thank the European Centre For Medium-Range Weather Forecasts for providing the ERA-Interim data.



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





**Figure 1.** Differences in surface air temperature (TAS, K) for (a) A-ERA-Interim, (b) $A_{PL}$-A, (c) $A4K_{PL4K}$-A4K, (d) $A4x_{PL4x}$-A4x, (e) $Arad4x_{PLrad4x}$-Arad4x and (f) $Asc_{PLsc}$-Asc. Equivalent differences between observations/simulations are given in (g)–(l) and (m)–(r) for pecipitation (PR, mm day$^{-1}$, CMAP data used in (g)) and mean sea level pressure (PSL, hPa, ERA-Interim data used in (m)), respectively. The points labelled with an "x" indicate the differences are statistically significant using the Student's t-test (p≤0.05).





**Figure 2.** Differences in surface air temperature (TAS, K) for (a) $[A4K_{PL4K}\text{-}A_{PL}]$–$[A4K\text{-}A]$ (b) $[A4x_{PL4x}\text{-}A_{PL}]$–$[A4x\text{-}A]$, (c) $[Arad4x_{PLrad4x}\text{-}A_{PL}]$–$[Arad4x\text{-}A]$ and (d) $[Asc_{PLsc}\text{-}A_{PL}]$–$[Asc\text{-}A]$. Equivalent differences between simulations are given in (e)–(h) and (i)–(l) for pecipitation (PR, mm day$^{-1}$) and mean sea level pressure (PSL, hPa), respectively. The points labelled with an "x" indicate the differences are statistically significant using the Student's t-test (p≤0.05).

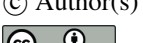



**Figure 3.** Differences (relative to A or $A_{PL}$—see key for each row) in global mean (column 1), ocean-only mean (column 2) and land-only mean (column 3) air temperature (K) for (a)–(c) the A4K experiments, (d)–(f) the A4x experiments, (g)–(i) Arad4x experiments and (j)–(l) the Asc experiments, respectively.



**Figure 4.** Differences in surface air temperature (TAS, K) for (a) A4K$_{PL}$-A$_{PL}$, (b) A4x$_{PL}$-A$_{PL}$, (c) Arad4x$_{PL}$-A$_{PL}$, (d) Asc$_{PL}$-A$_{PL}$, (e) A$_{PL4K}$-A$_{PL}$, (f) A$_{PL4x}$-A$_{PL}$, (g) A$_{PLrad4x}$-A$_{PL}$ and (h) A$_{PLsc}$-A$_{PL}$. The points labelled with an "x" indicate the differences are statistically significant using the Student's t-test (p≤0.05)



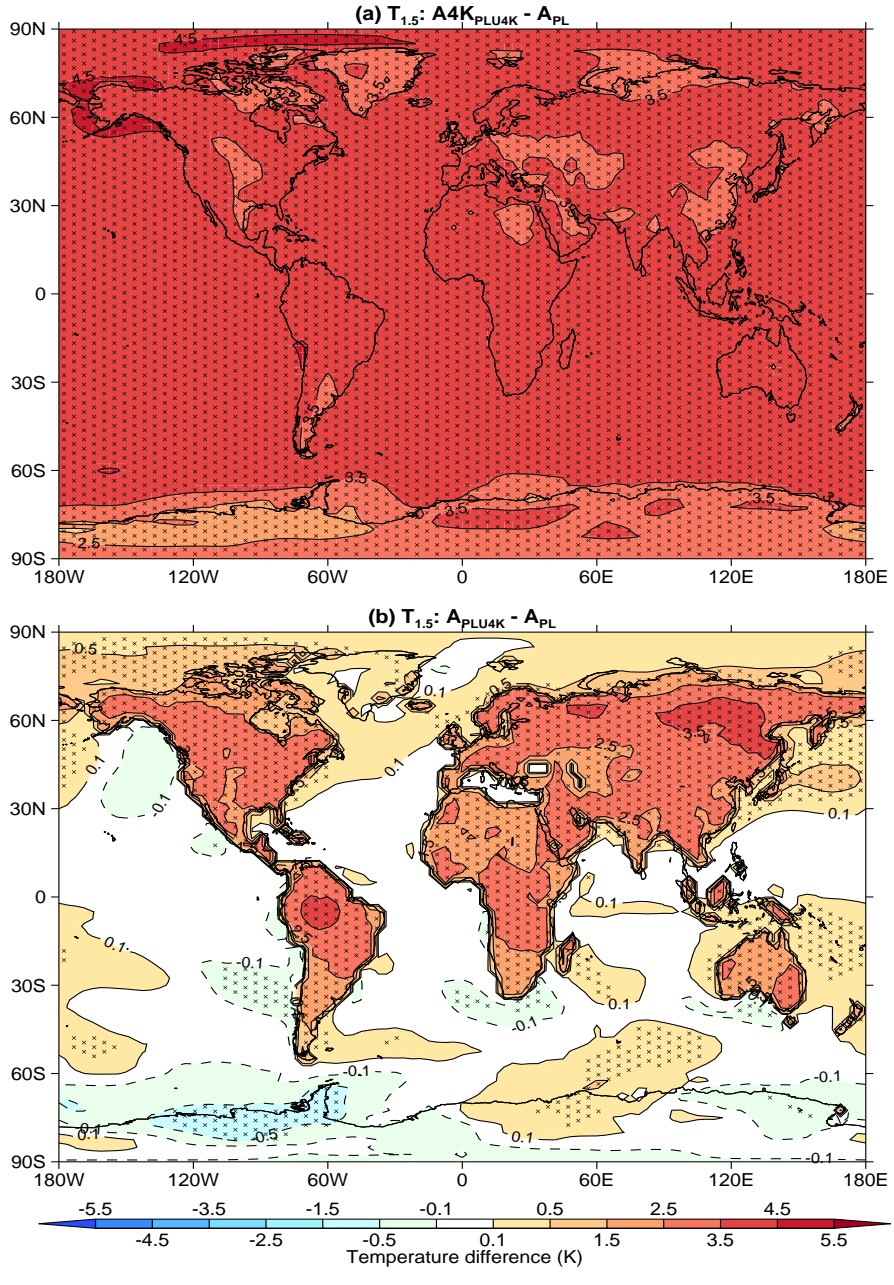

**Figure 5.** Differences in surface air temperature (TAS, K) for (a) A4K$_{PLU4K}$-A$_{PL}$, (b) A$_{PLU4K}$-A$_{PL}$. The points labelled with an "x" indicate the differences are statistically significant using the Student's t-test (p≤0.05)