# Peer review of "An ensemble of AMIP simulations with prescribed land surface temperatures"

_Geoscientific Model Development, 2018_

## Referee Comment (RC1) · Anonymous Referee #1 · 23 May 2018

This paper by Ackerley et al. describes a suite of fixed land temperature experiments with a single AGCM and provides a thorough validation of the experiment setup. The fixed land temperature experiments fill an important gap in the current model hierarchy, particularly in terms of understanding the traditional AMIP-style simulations. The paper shows that the land surface temperature can be prescribed in a way that is overall consistent with the free-land setup. These experiments, which are made publically available, therefore are of great scientific value. My only concern with this generally well-written paper is the lack of scientific analysis. While the main purpose of this paper is to provide a description and validation of experiment design, there are a few points that concerns the soundness of the experiments and should be better addressed. Particularly, the positive precipitation bias in the Amazon stands out as perhaps the biggest
caveat of the fixed land temperature experiments. If these experiments were to be used to study Amazon rainfall, such caveat needs to be better understood. And I suppose this paper should serve that purpose. The authors may expand on the hypothesis provided in a single sentence in L14-15 and elaborate on the mechanism provided in Cox et al. 1999.

Minor points / questions: 1. It might be worth mentioning the aquaplanet simulations that also have prescribed global surface temperature and have been used to indirectly study the impact of land surface temperature changes. For example, the CMIP6 standard aquaplanet simulations (e.g., He and Soden 2017) and the aquaplanet simulations with land-like temperatures (e.g., Tobias and Bjorn 2014). The lack of land in these aquaplanet simulations is an obvious shortcoming and the fixed land temperature experiments are a perfect solution. Tobias, B., and S. Bjorn, 2014: Climate and climate sensitivity to changing CO2 on an idealized land planet. J. Adv. Model. Earth Syst., 6, 1205–1223, doi:10.1002/2014MS000369. He, J., and B. J. Soden, 2017: A re-examination of the projected subtropical precipitation decline. Nat. Clim. Change, 7, 53–57, doi:10.1038/nclimate3157.

2. Page 4, Line 10. Are the land surface types prescribed or allowed to change?

3. Page 7, Line 3. How is the plant physiological effect switched off? Can it be explained in a couple of sentences?

4. Have the authors considered prescribing soil temperature and moisture separately (i.e., fix one and allow the other to change freely)?

---

## Referee Comment (RC2) · Anonymous Referee #2 · 25 May 2018

This paper describes an ensemble of experiments with the ACCESS AGCM in 'AMIP' style (with variants), but with the novel feature of fixing land surface temperatures. The method is described, then a large range of experiments where land surface temperatures are prescribed under various combinations of increased SSTs, fixed/variable stomatal resistance, changed CO2, changed solar constant. The analysis shows that the pattens of temperature, rainfall and MSLP are well preserved, and that the temperature fixing works effectively. Analysis is restricted to showing the effectiveness of the method throughout the very wide range of experiments.

This paper is a good fit for GMD, and is an effective and useful description of the methodology and effectiveness of this novel approach, and a generally clear description of the range of model results available for analysis.

I have only a few, mostly minor comments, essentially on clarifying the presentation:

(1) It is unclear what is done with snow cover, and surface temperatures over snow. What about inconsistencies such as snow cover present, but surface temperature > 0C? What if snow falls on an above zero point or is there already and T>0?

(2) Discuss early what is done over seaice (it is alluded to, but only later in the discussion).

(3) Why is soil moisture also set? This is not explained. Would it be possible to set T but not soil moisture? What would be the ramifications of that?

(4) P 3 lines 17-21: long complex sentence – break up

(5) P3, 25-27: Unclear sentence

(6) P4, l21: 'An initial estimate...'. Unclear – is this for freely varying T? Also start new para here.

(7) P7, l3: clarify why this is a 'radiative response' only: surface albedo of plants only is allowed to change? Also what about surface roughness?

(8) P7, L25: don't you mean on top of the ice sheets?

(9) Sometimes a hyphen means minus, and sometimes it means hyphen, and sometimes the mathematical minus is used. This all gets a bit confusing. Examples are P9, L19, 'A-ERA-Interim means 'A minus ERA-interim'. Fig 2 first line of caption, minus and dashed both mean minus. My suggestion (i) drop the hyphen e.g. in ERAinterim. (ii) when '-' means minus actually use the mathematical symbol (iii) the first time used, e.g. P9L19 spell it out "(i.e. A minus ERAinterim)".

(10) P9, L20. Please comment further on the temperature anomalies over the oceans, considering SSTs are fixed.

(11) P10, first para: worth noting that the biggest differences are over sea ice, where

surf temp is not prescribed.

(12) P10, L13: don't you mean that this is checking the climate change response is consistent for a given boundary change (e.g. SST)? After all you are comparing the differences.

(13) P11, L18: bias -> dry bias

(14) Table 4: why bother with italics AND asterisks?

(15) Section 3.4/Fig 3: why stop at 300hPa? It would be of interest to go higher, e.g. to 100hPa.

(16) P15, last line: spell out what a land/sea contrast of 0.25 means: Delta T ocean/Delta T land? See also line 9 on p16

(17) Fig 4. Can you put a heading over each column to help summarise for the reader? E.g. LHS could say "Land fixed, but changes to SST, CO2, stomates, SC", RHS could say "Land changes , isolated from prescribed land experiments" or similar. This would help the reader get their head around the complexity of these combinations.
* * *

---

## Author Comment (AC1) · 20 Jul 2018

**Response to Referee#1 on: "An ensemble of AMIP simulations with prescribed land surface temperatures" by Ackerley et al.**

**General Comments:**

*This paper by Ackerley et al. describes a suite of fixed land temperature experiments with a single AGCM and provides a thorough validation of the experiment setup. The fixed land temperature experiments fill an important gap in the current model hierarchy, particularly in terms of understanding the traditional AMIP-style simulations. The paper shows that the land surface temperature can be prescribed in a way that is over all consistent with the free-land setup. These experiments, which are made publically available, therefore are of great scientific value. My only concern with this generally well-written paper is the lack of scientific analysis. While the main purpose of this paper is to provide a description and validation of experiment design, there are a few points that concerns the soundness of the experiments and should be better addressed. Particularly, the positive precipitation bias in the Amazon stands out as perhaps the biggest caveat of the fixed land temperature experiments. If these experiments were to be used to study Amazon rainfall, such caveat needs to be better understood. And I suppose this paper should serve that purpose. The authors may expand on the hypothesis provided in a single sentence in L14-15 and elaborate on the mechanism provided in Cox et al. 1999.*

**Response:** The authors would like to thank the reviewer for undertaking their review of our work and for the positive and helpful comments that have been made. We do fully understand the reviewer's point about scientific analysis; however, we feel that providing a solid (and relatively simple) evaluation of the simulations we have undertaken is very important for others who wish to use these data. Providing in-depth scientific analysis of the runs would likely result in an extremely lengthy paper and make it difficult to simply present the biases associated with these experiments. This is why we made Geoscientific Model Development our publication choice, as it is the perfect platform for such a paper. It is hoped that others will provide more scientific insight (there are two papers the author is aware of that are in review and showcase some of the scientific value of these runs) by downloading the data themselves. The responses to the main points are given below.

The authors also fully agree with the need to provide more insight to the Amazon precipitation in the prescribed land simulations, but unfortunately we did not save the canopy water diagnostics as monthly means throughout the run and cannot re-run all the experiments (the lead author no longer works at the institute). Nevertheless, we did save the canopy water in the restart files (instantaneous values every 3 months, which gives 116 values to average over each run) and we have plotted the difference between $A_{PL}$ and A in Figure R1.1 below. There is higher canopy water storage in the $A_{PL}$ run relative to the A run, which matches the region of higher latent heat flux (Figure S3(a) in the supplementary material) and precipitation (Figure 1(h)) and supports our hypothesis in the main text. Given that this is an average over relatively few instantaneous values, the changes may not be highly statistically significant; however, the result supports the physical interpretation. We will add the figure to the supplementary material as further evidence of the stated process while also stating the associated caveats with the following text (the caption will be the same as that shown in Figure R1.1 below):

"The change in canopy water loading (kg m-2) taken from 116 instantaneous seasonal values (first day of each season over 29 years) from both the Apl and A simulations (Fig. S4). The 116

values are taken from the model restart files as climatological averages were not retained during the simulation. The positive change in canopy water loading corresponds with both the higher precipitation (Fig. 1(h) in the main text) and latent heat fluxes (Fig. S3(a)) that are seen in Apl relative to A, which agrees with the physical mechanism proposed in Section 3.2.1 of the main text."

**Specific comments:**

1. *It might be worth mentioning the aquaplanet simulations that also have prescribed global surface temperature and have been used to indirectly study the impact of land surface temperature changes. For example, the CMIP6 standard aquaplanet simulations (e.g., He and Soden 2017) and the aquaplanet simulations with land-like temperatures (e.g., Tobias and Bjorn 2014). The lack of land in these aquaplanet simulations is an obvious shortcoming and the fixed land temperature experiments are a perfect solution. Tobias, B., and S. Bjorn, 2014: Climate and climate sensitivity to changing CO2 on an idealized land planet. J. Adv. Model. Earth Syst., 6, 1205–1223, doi:10.1002/2014MS000369. He, J., and B. J. Soden, 2017: Are-examination of the projected subtropical precipitation decline. Nat. Clim. Change, 7, 53–57, doi:10.1038/nclimate3157.*

   **Response:** We have reviewed both of these pieces of work and agree with the reviewer that they are very useful additions to the cited literature in this paper. The He and Soden (2017) paper uses the opposite approach to our work in that an aquaplanet $4xCO_2$ simulation is compared with the standard AMIP $4xCO_2$ simulation. The impact of the land is inferred by subtracting one from the other (aqua vs AMIP with land) whereas our simulations suppress the land response (A4xpl) and compare it to the full (A4xpl4x) or land-only (Apl4x) simulations. Becker and Stevens (2014) on the other hand look at the response of various feedback processes in an idealised version of the ECHAM6 model. There are clear parallels between their idealised work and the simulations we have produced with prescribed land. Indeed, the cloud feedback processes highlighted by Becker and Stevens (2014) (e.g. the response of stratocumulus-type cloud to different land and $CO_2$ configurations) could be looked at in our simulations. We have therefore included references to both pieces of work in the 3rd paragraph of the Introduction.

2. *Page 4, Line 10. Are the land surface types prescribed or allowed to change?*

   **Response:** The land surface types do not change in the version of the model we have used (i.e. it is not an interactive scheme with respect to surface fractional type definition). They therefore do not change in any of the experiments given in the paper.

3. *Page 7, Line 3. How is the plant physiological effect switched off? Can it be explained in a couple of sentences?*

   **Response:** The calculation of photosynthesis in the vegetation scheme relies on the $CO_2$ concentration specified at the start of the run, as does the radiation scheme. In the 'rad' experiments, this is bypassed within the vegetation scheme by manually setting the $CO_2$

concentration to 346 ppmv in the code (whereas the atmosphere still 'sees' the specified value e.g. 1384 ppmv when quadrupled). This is how the physiological effect is 'switched off' i.e. it is a hard-wired bypassing of the value read in at the start of the run, which would normally be applied to both the vegetation and radiation schemes. We have therefore added:

"…and denoted as Arad4x from here on). This is done by setting the $CO_2$ concentration used in the photosynthesis calculation in the vegetation scheme to 346 ppmv but allowing the radiation scheme to 'see' the quadrupled value (i.e. 1384 ppmv)."

4. *Have the authors considered prescribing soil temperature and moisture separately (i.e., fix one and allow the other to change freely)?*

**Response:** The authors have considered such experiments; however, the aim of these experiments was to produce a set of AMIP prescribed land experiments that reproduced the free land experiments as closely as possible. Therefore, both the soil temperatures and moisture needed to be prescribed together to maintain consistency. This allowed us to initially identify any systematic errors that may have been present in the prescribed land runs. The soil temperature and moisture fields were then prescribed together in the individual forcing experiments to again remain consistent with the prescribed land experiments. This would then allow us to have confidence in whether the climatic changes seen in the simulations were 'land driven' (i.e. temperature and soil moisture) or not. Furthermore, by prescribing both fields together we are able to maintain the strong coupling between soil moisture and land temperature, segregating the two processes could be considered unphysical. Also, given the ~30 year length of the model simulations, there may be a long spin-up period required for the soil moisture field if it is allowed to vary freely with prescribed land temperatures, which could reduce the statistical significance of any observed changes in climate. The necessary testing for such an experiment was therefore beyond the scope and resources of the ensemble we have produced. Nevertheless, we would certainly advocate running such experiments in the future or encourage others to do so with the code that has been made available.

**Figure**

[Figure]

Figure R1.1: The difference in the canopy water loading (kg m$^{-2}$) in the A$_{PL}$ simulation relative to A.

---

## Author Comment (AC2) · 20 Jul 2018

**Response to Referee#2 on: "An ensemble of AMIP simulations with prescribed land surface temperatures" by Ackerley et al.**

**General comments:**

*This paper describes an ensemble of experiments with the ACCESS AGCM in 'AMIP' style (with variants), but with the novel feature of fixing land surface temperatures. The method is described, then a large range of experiments where land surface temperatures are prescribe under various combinations of increased SSTs, fixed/variable stomatal resistance, changed CO2, changed solar constant. The analysis shows that the pattens of temperature, rainfall and MSLP are well preserved, and that the temperature fixing works effectively. Analysis is restricted to showing the effectiveness of the method throughout the very wide range of experiments.*

*This paper is a good fit for GMD, and is an effective and useful description of the methodology and effectiveness of this novel approach, and a generally clear description of the range of model results available for analysis.*

**Response:** The authors would like to thank the reviewer for undertaking the review and for their positive and helpful comments. We have answered each of the points raised below and hope they are to the reviewer's satisfaction.

**Specific comments:**

*I have only a few, mostly minor comments, essentially on clarifying the presentation:*

**Response:** Please see our responses below.

1.  *It is unclear what is done with snow cover, and surface temperatures over snow. What about inconsistencies such as snow cover present, but surface temperature >0C? What if snow falls on an above zero point or is there already and T>0?*

    **Response:** The snow cover is allowed to change interactively in response to the prescribed surface temperature. Therefore (as would be the case in the 'free land' simulations), snow falling on land with temperatures above freezing would be melted based on the mass deposited and the melting rate at a given temperature. On surface points where snow is lying and the prescribed surface temperature rises above freezing point, the same process would occur (i.e. snow melt would occur based on the snow mass and surface temperature). There is always the possibility that some inconsistency may exist, which is unavoidable when running the model in such an idealised way. One option could have been to prescribe the snow field in these experiments too. Nevertheless, given the very small differences in the climatological fields presented (even at high latitudes where the snow issue would be largest) it is highly unlikely that there are any major inconsistencies that have a strong impact on the mean climate.

2.  *Discuss early what is done over sea ice (it is alluded to, but only later in the discussion).*

**Response:** We state that, "prescribed, observational SSTs and sea ice concentrations from 1979 to 2008 (30 years long)…" are used (Section 2.2.1). We do not prescribe the temperature of the sea ice and realise that this is not mentioned in the text. We have therefore adjusted the end of section 2.2.2 as follows to be clearer:

"Finally, land surface temperatures are not prescribed over both the permanent ice sheets (Antarctica and Greenland, to avoid the development of negative temperature biases that are discussed in Ackerley and Dommenget, 2016) and within/on sea ice. The impact of not specifying the land temperatures over the ice sheets or sea ice temperature is likely to be negligible (see Section 3)."

3. *Why is soil moisture also set? This is not explained. Would it be possible to set but not soil moisture? What would be the ramifications of that?*

**Response:** This point is also raised by the other reviewer (and also discussed there) and is an important point that should be addressed here too. Both the soil moisture and land temperature are set together to maintain as much consistency between the prescribed land and the 'free land' simulations. Segregating the two processes could be considered unphysical as the soil moisture and temperature fields are closely coupled. Another issue is associated with changing more than one variable at once. For example, in the $A_{4xPL}$ simulation, would precipitation changes be due to changing the land temperature or a change in soil moisture in response to the new temperature field? By prescribing both together we are able to say that the responses we see are due to the constrained land conditions as a whole, while accepting that we cannot separate this into the moisture-driven and temperature-driven effects. It is also worth noting that de-coupling the soil moisture and temperature fields (by allowing soil moisture to vary while prescribing temperature) would require a spin up period to allow the soil moisture to come into equilibrium with the temperature field (e.g. through evaporation). The atmosphere (e.g. precipitation) would then need time to reach an equilibrium with the new surface moisture field associated with the prescribed temperature field and then the soil moisture would need to adjust to the new precipitation field (and so on). This may therefore need multiple years of spin up, which would diminish the strength of the statistical relationships presented. Such an issue is not present in perturbed SST experiments as the ocean thermal properties are relatively uniform (relative to the land) but over the land, in order to maintain land surface temperatures and soil moisture consistency, it is therefore clear that both fields need to be prescribed.

While there are issues (see above) with allowing the soil moisture to change with prescribed temperatures, the authors do think that the work could be extended to undertake such experiments but it was beyond the scope of the current study. We state in Section 2.2.2 that, "The surface temperature, soil moisture and soil temperatures are all prescribed every 3 hours for the whole period 1980–2008 to minimise the differences between free and prescribed land simulations." Given the discussion above, the authors feel this is enough justification for the experiments that we have presented.

4. *P 3 lines 17-21: long complex sentence – break up*

**Response:** Sentence changed to: **"**All of the forcing agents outlined above have different impacts on land temperatures and both the global and regional climate. Therefore it would be useful to quantify the separate contributions of the land, the atmosphere (e.g. increased long-wave absorption), plant physiology and SSTs in the global and regional climate response."

5.  *P3, 25-27: Unclear sentence*

    **Response:** Lines 25-27 changed to: **"**The main aim of this study is to describe and validate a set of AMIP simulations against those with prescribed land conditions. This study also presents an evaluation of experiments that employ different combinations of land conditions with the different SST, CO2 and insolation specifications."

6.  *P4, l21: 'An initial estimate...'. Unclear – is this for freely varying T? Also start new para here.*

    **Response:** New paragraph started there and the sentence now begins with, "In the free land experiments…"

7.  *P7, l3:  clarify why this is a 'radiative response' only:  surface albedo of plants only is allowed to change? Also what about surface roughness?*

    **Response:** We have added the following to the end of point (4) in section 2.2.1:"…and denoted as Arad4x from here on). This is done by setting the $CO_2$ concentration used in the photosynthesis calculation in the vegetation scheme to 346 ppmv but allowing the radiation scheme to 'see' the quadrupled value (i.e. 1384 ppmv)."

    Plant albedo and roughness length do not change. The only variable that changes is the $CO_2$ concentration 'seen' by the radiation scheme while the vegetation scheme 'sees' the unperturbed value. This is in-line with the cited literature and that of CFMIP (Doutriaux-Boucher et al., 2009; Boucher et al., 2009; Andrews et al., 2011; Webb et al., 2017).

8.  *P7, L25: don't you mean on top of the ice sheets?*

    **Response:** Indeed. Changed to, "The impact of not specifying the temperatures over the ice sheets…".

9.  *Sometimes a hyphen means minus, and sometimes it means hyphen, and sometimes the mathematical minus is used. This all gets a bit confusing. Examples are P9, L19, 'A-ERA-Interim means 'A minus ERA-interim'.  Fig 2 first line of caption, minus and dashed both mean minus. My suggestion (i) drop the hyphen e.g. in ERAinterim. (ii) when '-' means minus actually use the mathematical symbol (iii) the first time used, e.g. P9L19 spell it out "(i.e. A minus ERAinterim)".*

**Response:** We can see the confusion caused by this. We have fixed it by doing the following:

- ERA-Interim is referred to as ERAI in the text now (except in the first instance).
- We have spelled out 'minus' instead of using the symbol to make things clearer.
- We have used '-' in Table 3 due to columns overrunning beyond the edge of the page (this can be resolved if the manuscript reaches the typesetting stage).
- We have not changed the figure notation (e.g. (a)–(d)) nor range notation (e.g. 1–3 K) as this is the journal's style.

10. *P9, L20. Please comment further on the temperature anomalies over the oceans, considering SSTs are fixed.*

    **Response:** There are several causes of the difference. One reason may be that ERA-Interim provides 2 m air temperatures whereas ACCESS outputs 1.5 m air temperature. The ACCESS air temperatures would therefore be systematically higher than those of ERA-Interim. Next, while the near surface air temperature will be constrained by the SST, the reanalysis data is adjusted with data assimilation whereas the ACCESS AMIP run is not. The difference may therefore be due to a systematic error in the lower atmosphere in the ACCESS run that is interpolated to the near surface whereas the ERA-Interim data are corrected towards observations (this point is made in section 4 of Kumar et al., 2013). The SST datasets are also different between ERA-Interim and those used by the ACCESS AMIP run (see Table 1 in Dee et al, 2011, and https://pcmdi.llnl.gov/mips/amip/amip2/#source_data) and the differences may be due to that. Overall, the figure is not given to highlight the differences between the ACCESS AMIP simulation and ERA-Interim, it is there to show that the differences in TAS between the prescribed and free land simulations are negligible when compared with the 'model biases' relative to observations. Please note however, we have changed the sentence in the paper to read 'Fig. 1(a)' as the original only said 'Fig. 1.'

11. *P10, first para: worth noting that the biggest differences are over sea ice, where surf temp is not prescribed.*

    **Response:** We have added the following to that paragraph: "…and the largest changes are at high-latitudes where sea ice is located (sea ice temperatures are not prescribed)."

12. *P10, L13: don't you mean that this is checking the climate change response is consistent for a given boundary change (e.g. SST)? After all you are comparing the differences.*

    **Response:** Agreed. We have changed the sentence to: "In order to validate whether the climate responses in the prescribed land simulations are consistent with their free land counterparts for a given boundary forcing…"

13. *P11, L18: bias -> dry bias*

**Response:** Changed as suggested.

14. *Table 4: why bother with italics AND asterisks?*

    **Response:** It makes the numbers stand out better (but still less conspicuous than the significant values) whereas just one or the other can make it difficult to distinguish at a glance. We prefer to do this rather than use bold type for the significant changes. The journal may decide to change it if we reach the typesetting stage and we would rather keep it if possible.

15. *Section 3.4/Fig 3: why stop at 300hPa? It would be of interest to go higher, e.g. to 100hPa.*

    **Response:** We have re-plotted Figure 3 to 100 hPa (as suggested by the reviewer) and it makes no difference to our statements/conclusions in the paper. We have included the new version that goes to 100 hPa in the revised version of the document. It is also included at the bottom of this response for reference.

16. *P15, last line: spell out what a land/sea contrast of 0.25 means: Delta T ocean/Delta T land? See also line 9 on p16.*

    **Response:** We have included a footnote at the first mention of the land/sea contrast that reads: "The global mean change in TAS over land divided by the global mean change in TAS over the ocean as done by Sutton et al. (2007)."

17. *Fig 4. Can you put a heading over each column to help summarise for the reader? E.g. LHS could say "Land fixed, but changes to SST, CO2, stomates, SC", RHS could say "Land changes, isolated from prescribed land experiments" or similar. This would help the reader get their head around the complexity of these combinations.*

    **Response:** We have included and annotation above each column with the left side saying, "AMIP land, perturbed boundary conditions" and the right side saying, "Land from perturbed boundary conditions, AMIP atmosphere/SST/solar constant".

**FIGURE**

[Figure]

**New version of Figure 3:** Differences (relative to A or AP L —see key for each row) in global mean (column 1), ocean-only mean (column 2) and land-only mean (column 3) air temperature (K) for (a)–(c) the A4K experiments, (d)–(f) the A4x experiments, (g)–(i) Arad4x experiments and (j)–(l) the Asc experiments, respectively.

**REFERNCES:**

Andrews, T., Doutriaux-Boucher, M., Boucher, O., and Forster, P. M.: A regional and global analysis of carbon dioxide physiological forcing and its impact on climate, Clim. Dynam., 36, 783–792, https://doi.org/10.1007/s00382-010-0742-1, 2011.

Boucher, O., Jones, A., and Betts, R. A.: Climate response to the physiological impact of carbon dioxide on plants in the Met Office Unified Model HadCM3, Clim. Dynam., 32, 237–249, https://doi.org/10.1007/s00382-008-0459-6, 2009.

Dee, D. P., Uppala, S. M., Simmons, A. J., Berrisford, P. , Poli, P. , Kobayashi, S. , Andrae, U. , Balmaseda, M. A., Balsamo, G. , Bauer, P. , Bechtold, P. , Beljaars, A. C., van de Berg, L. , Bidlot, J. , Bormann, N. , Delsol, C. , Dragani, R. , Fuentes, M. , Geer, A. J., Haimberger, L. , Healy, S. B., Hersbach, H. , Hólm, E. V., Isaksen, L. , Kållberg, P. , Köhler, M. , Matricardi, M. , McNally, A. P., Monge-Sanz, B. M., Morcrette, J. , Park, B. , Peubey, C. , de Rosnay, P. , Tavolato, C. , Thépaut, J. and Vitart, F. (2011), The ERA-Interim reanalysis: configuration and performance of the data assimilation system. Q.J.R. Meteorol. Soc., 137: 553-597. doi:10.1002/qj.828

Doutriaux-Boucher, M., Webb, M. J., Gregory, J. M., and Boucher, O.: Carbon dioxide induced stomatal closure increases radiative forcing via rapid reduction in low cloud, Geophys. Res. Lett., 36, L02 703, https://doi.org/10.1029/2008GL036273, 2009.

Kumar, A. , L. Zhang and W. Wang, 2013: Seas surface temperature-precipitation relationship in different reanalyses. Mon. Wea. Rev. 141, 1118-1123, https://doi.org/10.1175/MWR-D-12-00214.1

Webb, M. J., Andrews, T., Bodas-Salcedo, A., Bony, S., Bretherton, C. S., Chadwick, R., Chepfer, H., Douville, H., Good, P., Kay, J. E., Klein, S. A., Marchand, R., Medeiros, B., Siebesma, A. P., Skinner, C. B., Stevens, B., Tselioudis, G., Tsushima, Y., and Watanabe, M.: The Cloud Feedback Model Intercomparison Project (CFMIP) contribution to CMIP6, Geosci. Model Dev., 10, 359–384, https://doi.org/10.5194/gmd-10-359-2017, 2017.